# Detecting Out-of-distribution through the Lens of Neural Collapse

## Abstract

Out-of-Distribution (OOD) detection is essential for safe deployment; however, existing detectors exhibit generalization discrepancies and cost concerns. To address this, we propose a highly versatile and efficient OOD detector inspired by the trend of Neural Collapse on practical models, without requiring complete collapse. By analyzing this trend, we discover that features of in-distribution (ID) samples cluster closer to the weight vectors compared to features of OOD samples. Additionally, we reveal that ID features tend to expand in space to structure a simplex Equiangular Tight Framework, which explains the prevalent observation that ID features reside further from the origin than OOD features. Taking both insights from Neural Collapse into consideration, our OOD detector utilizes feature proximity to weight vectors and further complements this perspective by using feature norms to filter OOD samples. Extensive experiments on *off-the-shelf* models demonstrate the efficiency and effectiveness of our OOD detector across diverse classification tasks and model architectures, mitigating generalization discrepancies and improving *overall* performance.

## 1 Introduction

Machine learning models deployed in practice will inevitably encounter samples that deviate from the training distribution. As a classifier cannot make meaningful predictions on test samples that belong to classes unseen during training, it is important to actively detect and handle Out-of-Distribution (OOD) samples. Considering the diverse and oftentimes time-critical application scenarios, an OOD detector should be computationally efficient and can effectively generalize across various scenarios.

In this work, we focus on *post-hoc* methods, which address OOD detection independently of the training process. One line of prior work designs OOD scores over model output space (Djurisic et al., 2022; Hendrycks et al., 2019; Liang et al., 2018; Liu et al., 2020; Sun et al., 2021; Sun & Li, 2022) and another line of work focuses on the feature space, where OOD samples are observed to deviate from the clusters of ID samples (Lee et al., 2018; Mahalanobis, 2018; Sun et al., 2022; Tack et al., 2020). While existing research has made strides in OOD detection, they still face two major challenges: 1) maintaining detection effectiveness across different scenarios, and 2) ensuring computational efficiency for real-world deployment. For example, both output space and feature space methods suffer from performance discrepancy across different classification tasks, as shown in Table 1 (a). Specifically, strong algorithms on CIFAR-10 (Krizhevsky et al., 2009) OOD benchmarks perform suboptimally on ImageNet (Deng et al., 2009) OOD benchmarks, and vice versa. No existing algorithm can simultaneously rank in the top three across two benchmarks, leading to sub-optimal average performance as shown in Table 1 (b). Such discrepancy is also observed across different architectures, as shown in Table 2. In addition, feature space methods, which rely on auxiliary models, raise efficiency concerns. For example, Lee et al. (2018) learns a Gaussian mixture model from training features and detects OOD based on Mahalanobis distance Mahalanobis (2018); Sun et al. (2022) records the training features and measures OOD-ness based on the k-th nearest neighbor distance to the training features. As shown in Liu & Qin (2024), such reliance on auxiliary models introduces additional cost, posing challenges for time-critical applications.

To this end, we aim to develop an efficient and versatile OOD detector by focusing on the penultimate layer, i.e., the layer before the linear classification head. We take insights from *Neural Collapse* (Papyan et al., 2020), which characterizes the interplay between the linear classification head and the penultimate layer features in training. Neural Collapse is observed across diverse architectures

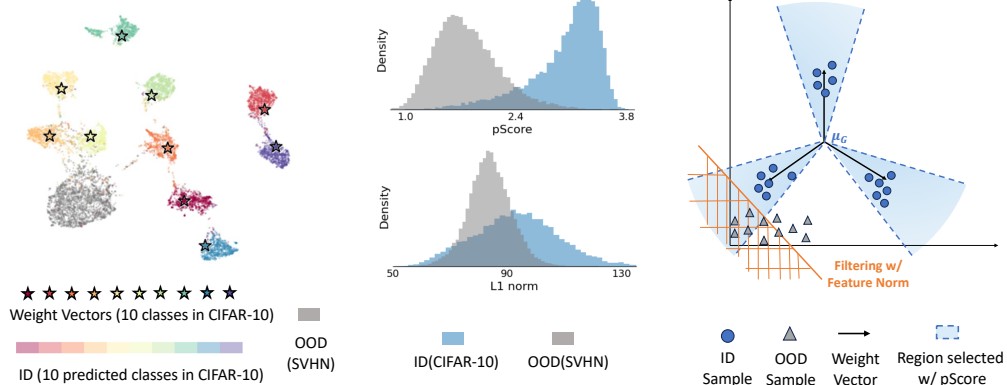

Figure 1: **Illustration of our framework inspired by Neural Collapse.** *Left:* On the penultimate layer, ID samples cluster near their predicted class weight vectors (marked by stars) while OOD samples reside separated, as shown by UMAP. *Middle:* ID and OOD samples are separated by pScore (Equation 6), which measures feature proximity to weight vectors. Also, ID samples tend to be further from the origin, illustrated with L1 norms. *Right:* ID samples cluster near a simplex Equiangular Tight Framework, illustrated with black arrows denoting weight vectors. We detect OOD by thresholding on pScore, selecting blue-shaded hypercones centered at weight vectors, with OOD samples outside these areas. We also filter OOD samples characterized by smaller feature norms. *Left & Middle* present a CIFAR-10 ResNet-18 classifier with OOD set SVHN. While Neural Collapse does not **completely converge** (*Left*), the ID/OOD relationship inspired by the **trend** remains valid (*Middle*). *Right* depicts our scheme on a three-class classifier with 2D penultimate space.

and classification tasks (see Appendix E). While the complete collapse requires strict conditions like prolonged training, we leverage its early-stage *trend* observed in (He & Su, 2023) to study practical models. The effectiveness of prior methods utilizing Neural Collapse in OOD detection Zhang et al. (2022); Ammar et al. (2023) further supports the prevalence of such trend in practical models.

Particularly, we revisit the observation that ID features tend to form clusters while OOD features reside apart. While this observation is well-established in prior literature Lee et al. (2018); Sun et al. (2022); Tack et al. (2020), the underlying mechanism remains largely unexplained. Separately, Neural Collapse reveals that features of each class gradually converge toward a single point during training. We suggest that the clustering behavior observed in *off-the-shelf* models can reflect the trend of Neural Collapse. Inspired by this, we leverage the landscape of Neural Collapse to study:

*Where do features of ID samples form clusters?*

To address the question, we first demonstrate that as a deterministic effect of Neural Collapse, features of training samples will converge towards *the weight vectors of the predicted class*. Additionally, Neural Collapse reveals that training features also converge towards a simplex Equiangular Tight Framework (ETF) (Equation 1). The spatial structure of an ETF, illustrated in Figure 1 *Right*, corresponds to the maximum separation in space achievable by equiangular vectors, requiring the features to reside sufficiently far from the origin.

The complete convergence landscape of Neural Collapse sheds light on the geometric structure of ID clusters on practical models. Specifically, for ID test samples, drawn from the same distribution as training samples, we anticipate a similar *trend* of clustering behavior towards the weight vectors and towards an ETF. Conversely, OOD samples do not undergo the same training process, which enables the model to align features with weight vectors and to expand features to accommodate the spatial structure of ETF in Neural Collapse. Therefore, we do not expect the model to effectively align the weight vectors learned from ID features with unseen OOD features. Nor do we anticipate the model to posit OOD features far from the origin to structure an

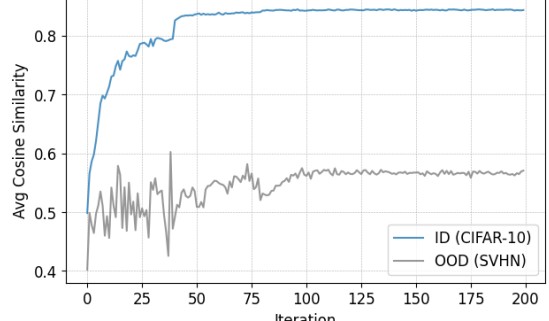

Figure 2: **Features of ID samples tend to cluster closer to the predicted class weight vectors**, indicated by higher average cosine similarity (Equation 5) than OOD. This observation, inspired by the trend of Neural Collapse, emerges early in the training of this CIFAR-10 ResNet-18 classifier, with OOD set SVHN, without requiring convergence.

ETF. To validate our hypotheses, we trace a model's training stages in Figure 2. We observe that ID samples consistently cluster closer to the weight vectors than OOD samples. This observation emerges early during training, without requiring complete convergence of Neural Collapse. Our observation is reinforced in the UMAP (McInnes et al., 2018) visualization on an *off-the-shelf* CIFAR-10 classifier with ResNet-18 backbone in Figure 1 Left. Here, ID features do not completely collapse into weight vectors. Nevertheless, ID features cluster near predicted class weight vectors (marked by stars), whereas OOD features are distant. Combining our observation with (Zhu et al., 2021), which show the weight vectors form an ETF, we conclude that ID features are driven to structure the ETF during training, whereas OOD features lack the incentive to expand in space to form an ETF. Note that the lack of incentive for OOD features to expand explains the well-established observation (Tack et al., 2020; Huang et al., 2021; Sun et al., 2022) that OOD features tend to reside closer to the origin, offering an alternative to model confidence view in Park et al. (2023).

Based on our understanding, we design an efficient and versatile OOD detector. We first leverage feature proximity to the weight vectors to characterize ID clustering, bypassing auxiliary models and reducing the computational cost. Specifically, we define an angle-based proximity score as the norm of the projection of the weight vector of the predicted class onto the sample feature. As shown in Figure 1 *Middle*, our proximity score can effectively separate ID/OOD. A higher score indicates closer proximity and a lower chance of OOD-ness. Geometrically, thresholding on the score selects hyper-cones centered at the weight vector, as illustrated in Figure 1 *Right*. Notably, our proximity score effectively incorporates class-specific information and brings in performance benefits as well as efficiency gain. Complementing the proximity score's contingency on ID clustering, we also consider feature distance to the origin. Specifically, ID features tend to reside further from the origin as they expand in space to form an ETF, whereas OOD features tend to reside near the origin, as illustrated by Figure 1 *Right*. Using the L1 norm as an example metric for distance to the origin, we observe that ID features can be separated from OOD features, as supported by Figure 1 *Middle*. Combining both aspects, we propose **N**eural **C**ollapse **I**nspired OOD Detector (`NCI`).

Notably, prior methods, e.g., KNN Sun et al. (2022), focus on ID clustering but do not explicitly consider feature distance to the origin. Such approaches fall short in scenarios like ImageNet benchmarks but yield superior performance in CIFAR-10 benchmarks in Table 1a. Conversely, methods such as Energy Liu et al. (2020), Energy-based ASH Djurisic et al. (2022), and, Energy-based Scale Xu et al. (2023) inherently utilize feature distance to the origin by considering log-sum-exp of logits, yet largely overlook ID clustering. These approaches excel in scenarios like ImageNet, but perform sub-optimally in others, e.g., CIFAR-10. Through the lens of Neural Collapse, we explain, connect, and complete prior methods under a holistic view, resulting in reduced latency and generalization discrepancies.

We summarize our main contributions below:

- **Understanding and Observation:** By analyzing ID clustering through the trend of Neural Collapse, we novelly establish the significance of weight vectors in the clusters. We also explain the observation that ID features tend to be farther from the origin from a spatial structure perspective. Our understanding and observation do not depend on complete complete Neural Collapse convergence.

- **OOD Detector:** We leverage feature proximity to the weight vectors of predicted classes for OOD detection, integrating class-specific information. Complementary to feature clustering, we propose to detect OOD samples by thresholding the feature distance to the origin.

- **Experimental Analysis:** We evaluate `NCI` across diverse classification tasks (CIFAR-10, CIFAR-100, ImageNet) and model architectures (ResNet, DenseNet Huang et al. (2017), ViT Dosovitskiy et al. (2020), Swin Liu et al. (2022)). Rather than focusing on *individual* benchmarks, `NCI` reduces the generalization discrepancies and improves the *overall* effectiveness. In addition, `NCI` matches the latency of vanilla *softmax-confidence* detector.

**Remark on Convergence of Neural Collapse & NCI Effectiveness.** *Complete* convergence of Neural Collapse for ID samples often requires strict conditions unmet in practice. However, NCI does not depend on convergence; instead, it leverages the *trend* of Neural Collapse, which we empirically validate on practical models. Additionally, the effectiveness of prior methods utilizing Neural Collapse in OOD detection (Zhang et al., 2022; Ammar et al., 2023) further supports the prevalence of the trend of Neural Collapse in practical models. We extensively validated the effectiveness of NCI on practical models without convergence requirement.

**Remark on NCI Performance.** NCI does not focus on individual benchmarks. While NCI may not achieve the best performance on every benchmark, existing detectors exhibit larger generalization discrepancies. NCI mitigates the discrepancies, achieving the best *overall* performance across all benchmarks. Additionally, NCI incurs minimal latency and enhances computational efficiency.

## 2 PROBLEM SETTING

We consider a data space $\mathcal{X}$, a class set $\mathcal{C}$, and a classifier $f : \mathcal{X} \to \mathcal{C}$, which is trained on samples *i.i.d.* drawn from joint distribution $\mathbb{P}_{\mathcal{X}\mathcal{C}}$. We denote the marginal distribution of $\mathbb{P}_{\mathcal{X}\mathcal{C}}$ on $\mathcal{X}$ as $\mathbb{P}^{in}$. And samples drawn from $\mathbb{P}^{in}$ are In-Distribution (ID) samples. In practice, the classifier $f$ may encounter $\boldsymbol{x} \in \mathcal{X}$ yet is not drawn from $\mathbb{P}^{in}$. We say such samples are Out-of-Distribution (OOD).

In this work, we focus on detecting OOD samples from *classes unseen during training*, for which the classifiers cannot make meaningful predictions. The OOD detector $D : \mathcal{X} \to \{\text{ID}, \text{OOD}\}$ is commonly constructed as: $D(\boldsymbol{x}) = \begin{cases} \text{ID} & \text{if } s(\boldsymbol{x}) \geq \tau \\ \text{OOD} & \text{if } s(\boldsymbol{x}) < \tau \end{cases}$, where $s : \mathcal{X} \to \mathbb{R}$ is a score function of design and $\tau$ is the threshold. Considering the diverse application scenarios, an ideal OOD detector should be efficient and generalizable. In this work, we leverage insights from Neural Collapse to achieve reduced computational costs and minimize generalization discrepancies.

## 3 OOD DETECTION THROUGH THE LENS OF NEURAL COLLAPSE

In this section, we re-examine the observation in Lee et al. (2018); Sun et al. (2022) that ID features tend to form clusters while OOD features deviate from the clusters. We suggest understanding the clustering phenomenon can reflect the trend of the Neural Collapse (Papyan et al., 2020), which does not necessitate complete Neural Collapse convergence. Leveraging the landscape revealed by Neural Collapse, we examine:

*Where do features of ID samples form clusters?*

Through analytical and empirical study, we hypothesize and validate with pre-trained models that (1) ID features tend to cluster closer to the weight vectors compared to OOD features; (2) ID clusters tend to reside further from the origin, as necessitated by their spatial structure. From our understanding, we develop a *post-hoc* OOD detector with enhanced efficiency and effectiveness.

### 3.1 NEURAL COLLAPSE: CONVERGENCE OF TRAINING FEATURES

Neural Collapse, first observed in Papyan et al. (2020), occurs on the penultimate layer across canonical classification settings. To formally introduce the concept, we use $\boldsymbol{h}_{i,c}$ to denote the penultimate layer feature of the $i_{th}$ training sample with ground truth / predicted label $c$, Neural Collapse is framed in relation to

- the feature global mean, $\boldsymbol{\mu}_G = \text{Ave}_{i,c} \boldsymbol{h}_{i,c}$, where Ave is the average operation;
- the feature class means, $\boldsymbol{\mu}_c = \text{Ave}_i \boldsymbol{h}_{i,c}, \; \forall c \in \mathcal{C}$;
- the within-class covariance, $\boldsymbol{\Sigma}_W = \text{Ave}_{i,c} (\boldsymbol{h}_{i,c} - \boldsymbol{\mu}_c)(\boldsymbol{h}_{i,c} - \boldsymbol{\mu}_c)^T$;
- the between-class covariance, $\boldsymbol{\Sigma}_B = \text{Ave}_c (\boldsymbol{\mu}_c - \boldsymbol{\mu}_G)(\boldsymbol{\mu}_c - \boldsymbol{\mu}_G)^T$;
- the linear classification head, i.e. the last layer of the NN, $\arg\max_{c \in \mathcal{C}} \boldsymbol{w}_c^T \boldsymbol{h} + b_c$, where $\boldsymbol{w}_c$ and $b_c$ are parameters corresponding to class $c$.

Neural Collapse comprises four inter-related limiting behaviors:

**(NC1) Within-class variability collapse:** $\boldsymbol{\Sigma}_W \to \boldsymbol{0}$

**(NC2) Convergence to a simplex Equiangular Tight Frame (ETF):**
$$\big| \|\boldsymbol{\mu}_c - \boldsymbol{\mu}_G\|_2 - \|\boldsymbol{\mu}_{c'} - \boldsymbol{\mu}_G\|_2 \big| \to 0, \; \forall \, c, \; c'$$
$$\frac{(\boldsymbol{\mu}_c - \boldsymbol{\mu}_G)^T (\boldsymbol{\mu}_{c'} - \boldsymbol{\mu}_G)}{\|\boldsymbol{\mu}_c - \boldsymbol{\mu}_G\|_2 \|\boldsymbol{\mu}_{c'} - \boldsymbol{\mu}_G\|_2} \to \frac{|\mathcal{C}|}{|\mathcal{C}| - 1} \delta_{c,c'} - \frac{1}{|\mathcal{C}| - 1} \tag{1}$$

where $\delta_{c,c'}$ is the Kronecker delta symbol.

**(NC3) Convergence to self-duality:**
$$\frac{\boldsymbol{w}_c}{\|\boldsymbol{w}_c\|_2} - \frac{\boldsymbol{\mu}_c - \boldsymbol{\mu}_G}{\|\boldsymbol{\mu}_c - \boldsymbol{\mu}_G\|_2} \to \boldsymbol{0}$$

**(NC4) Simplification to nearest class center:**

$$\arg\max_{c\in\mathcal{C}} \boldsymbol{w}_c^T \boldsymbol{h} + b_c \rightarrow arg\min_{c\in\mathcal{C}} \|\boldsymbol{h} - \boldsymbol{\mu}_c\|_2$$

We first remark on **(NC2)** that an ETF achieves the maximum separation possible for globally centered equiangular vectors Papyan et al. (2020) and extends in space, as visualized in Figure 1 *Right*. Since training features converge towards an ETF, they need to have sufficient norms to accommodate the spatial arrangement.

We next build on **(NC1)** and **(NC3)** to demonstrate in the following that training features converge towards the weight vectors of the linear classification head, up to a scaling factor.

**Theorem 3.1.** *(NC1) and (NC3) imply that for any sample $i$ and its predicted class $c$, we have*

$$(\boldsymbol{h}_{i,c} - \boldsymbol{\mu}_G) \rightarrow \lambda \boldsymbol{w}_c \tag{2}$$

*in the Terminal Phase of Training, where $\lambda = \dfrac{\|\boldsymbol{\mu}_c - \boldsymbol{\mu}_G\|_2}{\|\boldsymbol{w}_c\|_2}$.*

*Proof.* Considering that $(\boldsymbol{h}_{i,c} - \boldsymbol{\mu}_c)(\boldsymbol{h}_{i,c} - \boldsymbol{\mu}_c)^T$ is positive semi-definite for any $i$ and $c$. $\boldsymbol{\Sigma}_W \rightarrow \boldsymbol{0}$ thus implies $(\boldsymbol{h}_{i,c} - \boldsymbol{\mu}_c)(\boldsymbol{h}_{i,c} - \boldsymbol{\mu}_c)^T \rightarrow \boldsymbol{0}$ and $\boldsymbol{h}_{i,c} - \boldsymbol{\mu}_c \rightarrow \boldsymbol{0}$, $\forall i, c$. With algebraic manipulations, we have

$$\frac{\boldsymbol{h}_{i,c} - \boldsymbol{\mu}_G}{\|\boldsymbol{\mu}_c - \boldsymbol{\mu}_G\|_2} - \frac{\boldsymbol{\mu}_c - \boldsymbol{\mu}_G}{\|\boldsymbol{\mu}_c - \boldsymbol{\mu}_G\|_2} \rightarrow \boldsymbol{0}, \ \forall i, c \tag{3}$$

Applying the triangle inequality, we have

$$\left|\frac{\boldsymbol{h}_{i,c} - \boldsymbol{\mu}_G}{\|\boldsymbol{\mu}_c - \boldsymbol{\mu}_G\|_2} - \frac{\boldsymbol{w}_c}{\|\boldsymbol{w}_c\|_2}\right| \leq \left|\frac{\boldsymbol{h}_{i,c} - \boldsymbol{\mu}_G}{\|\boldsymbol{\mu}_c - \boldsymbol{\mu}_G\|_2} - \frac{\boldsymbol{\mu}_c - \boldsymbol{\mu}_G}{\|\boldsymbol{\mu}_c - \boldsymbol{\mu}_G\|_2}\right| + \left|\frac{\boldsymbol{w}_c}{\|\boldsymbol{w}_c\|_2} - \frac{\boldsymbol{\mu}_c - \boldsymbol{\mu}_G}{\|\boldsymbol{\mu}_c - \boldsymbol{\mu}_G\|_2}\right|. \tag{4}$$

Since both terms on the RHS converge to $\boldsymbol{0}$, as demonstrated by equation 3 and **(NC3)**, it follows that the LHS also converges to $\boldsymbol{0}$. $\qquad\square$

## 3.2 Trend of Neural Collapse and Geometric Structure of the ID Clusters

While the complete collapse occurs during the Terminal Phase of Training (TPT) where training error vanishes and the training loss is trained towards zero, it is observed in He & Su (2023) that the trend of Neural Collapse establishes in the early stages of training. We thus suggest that the clustering behavior of ID features observed in *off-the-shelf* models can reflect a trend of Neural Collapse, corresponding to the within-class variability collapse **(NC1)**. Such a trend does not necessitate a complete convergence and the prevalence of the trend in practical models is supported by the effectiveness of prior OOD detector which leverages Neural Collapse (Zhang et al., 2022; Ammar et al., 2023). In light of this, we leverage the landscape of Neural Collapse revealed in Theorem 3.1 and **(NC2)** to examine the geometry of ID feature clusters.

Since ID test samples are drawn from the same distribution as the training samples, we anticipate a similar pattern in their features. Specifically, we expect ID features to cluster towards the weight vectors of their predicted class during training. Additionally, we expect ID features to reside near a simplex Equiangular Tight Frame (ETF), thereby acquiring sufficient norm. Conversely, OOD samples are unseen during training and do not undergo the process of iterative adjustment, which drives the Neural Collapse phenomenon. Thus we expect the model to be less effective in aligning the OOD samples with weight vectors, placing OOD further from the weight vectors than ID features. Meanwhile, we do not expect the model to effectively align the OOD samples with an ETF.

In Figure 2, we validate our hypothesis across the training process of a CIFAR-10 classifier with ResNet-18 backbone. In Figure 2, we compute over the ID set (CIFAR-10) and OOD set (SVHN) the average cosine similarity between the centered feature $\boldsymbol{h}_i - \boldsymbol{\mu}_G$ and the weight vector $\boldsymbol{w}_c$ of the predicted class $c$, i.e.,

$$Avg_i \ \frac{(\boldsymbol{h}_i - \boldsymbol{\mu}_G) \cdot \boldsymbol{w}_c}{\|\boldsymbol{h}_i - \boldsymbol{\mu}_G\|_2 \|\boldsymbol{w}_c\|_2} \tag{5}$$

We observe that ID features have higher similarity scores and cluster closer to the weight vectors than OOD features. This relative relationship emerges early in training, without requiring full convergence. We further reinforce our observation in Figure 1 *Left* where we visualize ID features, OOD features, and weight vectors of a CIFAR-10 classifier with UMAP(McInnes et al., 2018). ID features are color-coded to align with the weight vectors (marked by stars) of their predicted classes,

revealing a distinct clustering pattern near the weight vectors. Conversely, OOD features reside further away. While ID features don't fully collapse onto weight vectors, showing incomplete Neural Collapse, the emerging trend still holds, and the ID/OOD relationship remains valid.

Additionally, we combine our observation with (Zhu et al., 2021), showing that the weight vectors form an ETF during training. Our observed proximity to the weight vectors thus also validates the clustering of ID features near an ETF and the divergence of OOD from this structure. The lack of structure and incentives to extend in space explains the relatively smaller norm of OOD features.

### 3.3 OUT-OF-DISTRIBUTION DETECTION

Based on our understanding, we design an efficient and versatile OOD detector. Specifically, we propose to detect OOD based on feature proximity to the weight vectors of the predicted class. For the proximity metric, we avoid Euclidean-based metrics as they require estimating the scaling factor $\lambda$ in Equation 2. This estimation tends to be imprecise for general classifiers which may cease training prior to convergence, resulting in suboptimal performance of Euclidean-based metrics shown in Appendix B. Instead, we design an angle-based metric, adjusted for class-wise difference. Specifically, we propose to quantify the proximity as the norm of projection of the weight vector $\boldsymbol{w}_c$ onto the centered feature $\boldsymbol{h} - \boldsymbol{\mu}_G$, where $c$ corresponds to the predicted class, i.e.,

$$\texttt{pScore} = cos(\boldsymbol{w}_c, \boldsymbol{h} - \boldsymbol{\mu}_G)\|\boldsymbol{w}_c\|_2, \tag{6}$$

where $cos(\boldsymbol{w}_c, \boldsymbol{h} - \boldsymbol{\mu}_G) = \frac{(\boldsymbol{h}-\boldsymbol{\mu}_G)\cdot\boldsymbol{w}_c}{\|\boldsymbol{h}-\boldsymbol{\mu}_G\|_2\|\boldsymbol{w}_c\|_2}$. A higher $\texttt{pScore}$ indicates closer proximity to the weight vector and thus a lower chance of OOD-ness. Geometrically, thresholding on $\texttt{pScore}$ selects infinite hyper-cones centered at the weight vectors, as illustrated in Figure 1 *Right*. Within the same predicted class, $\texttt{pScore}$ is proportional to the cosine similarity. Across different classes, $\texttt{pScore}$ adapts to class-wise difference by selecting wider hyper-cones for classes with larger weight vectors, which tend to have larger decision regions. As shown in Appendix B, our $\texttt{pScore}$ with class-wise adjustment outperforms vanilla cosine similarity. Notably, our $\texttt{pScore}$ incorporates class-specific information into characterizing ID clustering by using the weight vectors of the predicted class. This brings in additional gain in detection effectiveness, as we shall see in Section 4.

While $\texttt{pScore}$ enhances efficiency and effectiveness, its performance is intrinsically contingent on the strength of ID clustering. Such contingency, widely exhibited by clustering-based methods Lee et al. (2018); Sun et al. (2022); Tack et al. (2020), poses challenges on classifiers with less pronounced ID clustering, such as ImageNet ResNet-50 in Section 4.1. To mitigate such discrepancy, we complement $\texttt{pScore}$ by considering the distance of ID clusters to the origin. Specifically, we enhance our proximity score by incorporating feature norms to filter out OOD near the origin, as illustrated in Figure 1 *Right*. Taking L1 norm as an example, we define our detection score as $\texttt{pScore} + \alpha\|\boldsymbol{h}\|_1$, where $\alpha$ controls the filtering strength and can be selected from a validation set as detailed in Section 4. We refer readers to Section 4.3 for the effect of different orders of $p$-norm. Thresholding on the detection score, we have **N**eural **C**ollapse **I**nspired OOD Detector ($\texttt{NCI}$): A lower score indicates a higher chance of OOD-ness.

$\texttt{NCI}$ has $O(P)$ complexity, where $P$ is the penultimate layer dimension. The complexity theoretically ensures computational scalability of $\texttt{NCI}$ on large models. Empirically, $\texttt{NCI}$ maintains inference latency comparable to the vanilla *softmax-confidence* detector, as we shall see in Section 4.

## 4 EXPERIMENTS

In this section, we extensively evaluate $\texttt{NCI}$ across classification tasks: CIFAR-10, CIFAR-100 (see App. D), ImageNet, as well as model architectures: ResNet, DenseNet (see App. D), ViT, Swin. We compare $\texttt{NCI}$ against *thirteen* baseline methods. While $\texttt{NCI}$ may not achieve the best performance on individual benchmarks, it mitigates the exisitng generalization discrepancies and achieves the best **overall** performance with **minimal** latency. Following the OpenOOD benchmark Zhang et al. (2023), we evaluate on *six* OOD sets for CIFAR-10 and CIFAR-100 classifiers and *five* for ImageNet classifiers. Performance is evaluated using two widely recognized metrics: the False Positive Rate at 95% True Positive Rate (FPR95) and the Area Under the Receiver Operating Characteristic Curve (AUROC). Lower FPR95 and higher AUROC values indicate better performance. We also report the per-image inference latency (in milliseconds) evaluated on a Tesla T4 GPU. In our experiments, other than the ablation study in Section 4.3, we use the $L1$-norm as the filtering term and select the filtering strength $\alpha$ from $\{10^{-4}, 10^{-3}, 10^{-2}, 10^{-1}\}$ based on a validation set generated per pixel

Table 1: `NCI` reduces discrepencies and improves **overall performance** on CIFAR-10 and ImageNet benchmarks with **minimal latency**. CIFAR-10 uses ResNet-18 and ImageNet uses ResNet-50.

| | Methods | CIFAR-10 OpenOOD Benchmark | | | | | | | ImageNet OpenOOD Benchmark | | | | | |
|---|---|---|---|---|---|---|---|---|---|---|---|---|---|---|
| | | CIFAR-100 | TIN | MNIST | SVHN | Texture | Place365 | AVG | SSB-hard | NINCO | iNaturalist | Texture | OpenImage-O | AVG |
| | | *Evaluation under FPR95* ↓ | | | | | | | | | | | | |
| | MSP * | 53.08 | 43.27 | 23.64 | 25.82 | 34.96 | 42.47 | 37.20 | 74.49 | 56.88 | 43.34 | 60.87 | 50.13 | 57.14 |
| CIFAR-10 Strong | ODIN | 77.00 | 75.38 | 23.83 | 68.61 | 67.70 | 70.36 | 63.81 | 76.83 | 68.16 | 35.96 | 49.24 | 46.67 | 55.38 |
| CIFAR-10 Strong | Energy * | 66.60 | 56.08 | 24.99 | 35.12 | 51.82 | 54.85 | 48.24 | 76.54 | 60.58 | 31.30 | 45.77 | 38.09 | 50.46 |
| CIFAR-10 Strong | MDS | 91.87 | 92.66 | 1.30 | 74.34 | 76.07 | 94.16 | 38.11 | 95.19 | 91.86 | 84.23 | 73.31 | 90.77 | 87.07 |
| CIFAR-10 Strong | KNN | 37.64 | 30.37 | 20.05 | 22.60 | 24.06 | 30.38 | **27.38** | 83.36 | 58.39 | 40.80 | 17.31 | 44.27 | 48.82 |
| CIFAR-10 Strong | ViM | 49.19 | 40.49 | 18.36 | 19.29 | 21.14 | 41.43 | 31.65 | 80.41 | 62.29 | 30.68 | 10.51 | 32.82 | 43.34 |
| CIFAR-10 Strong | fDBD * | 61.36 | 49.04 | 37.44 | 56.99 | 36.30 | 32.11 | 45.54 | 77.28 | 52.13 | 22.00 | 36.05 | 29.94 | 43.48 |
| ImageNet Strong | GradNorm | 94.54 | 94.89 | 85.41 | 91.65 | 98.09 | 92.46 | 92.84 | 78.24 | 79.54 | 32.03 | 43.27 | 68.46 | 60.31 |
| ImageNet Strong | NECO | 72.63 | 62.47 | 15.52 | 30.16 | 60.12 | 56.08 | 49.50 | 72.31 | 55.08 | 27.05 | 48.25 | 33.42 | 47.22 |
| ImageNet Strong | ReAct | 67.40 | 59.71 | 33.77 | 50.23 | 51.42 | 44.20 | 51.12 | 77.55 | 55.82 | 16.72 | 29.64 | 32.58 | 42.46 |
| ImageNet Strong | DICE | 73.71 | 66.37 | 30.83 | 36.61 | 62.42 | 77.19 | 57.85 | 77.96 | 66.90 | 33.37 | 44.28 | 47.83 | 52.03 |
| ImageNet Strong | ASH | 87.31 | 86.25 | 70.00 | 83.64 | 84.59 | 77.89 | 81.61 | 73.66 | 52.97 | 14.04 | 15.26 | 29.15 | 37.02 |
| ImageNet Strong | Scale | 86.10 | 83.64 | 35.12 | 68.59 | 84.90 | 66.14 | 70.75 | 67.72 | 51.86 | 9.52 | 17.51 | 28.17 | **34.96** |
| | NCI w/o filter* | 51.92 | 43.54 | 32.63 | 28.92 | 26.53 | 34.01 | 36.26 | 82.14 | 56.41 | 24.11 | 34.32 | 30.94 | 45.58 |
| | NCI | 51.83 | 43.60 | 32.64 | 29.01 | 26.54 | 33.99 | 36.27 | 73.29 | 53.86 | 14.31 | 23.79 | 30.98 | 39.25 |
| | | *Evaluation under AUROC* ↑ | | | | | | | | | | | | |
| | MSP * | 87.19 | 88.87 | 92.63 | 91.46 | 89.89 | 88.92 | 89.83 | 72.09 | 79.95 | 88.41 | 82.43 | 84.86 | 81.55 |
| CIFAR-10 Strong | ODIN | 82.18 | 83.55 | 95.24 | 84.58 | 86.94 | 85.07 | 86.26 | 71.74 | 77.77 | 91.17 | 89.00 | 88.23 | 83.58 |
| CIFAR-10 Strong | Energy * | 86.36 | 88.80 | 94.32 | 91.79 | 89.47 | 89.25 | 90.00 | 72.08 | 79.70 | 90.63 | 88.70 | 89.06 | 84.04 |
| CIFAR-10 Strong | MDS | 61.29 | 59.57 | 99.17 | 66.56 | 77.40 | 52.47 | 69.41 | 43.92 | 55.41 | 61.82 | 79.94 | 60.80 | 60.38 |
| CIFAR-10 Strong | KNN | 89.73 | 91.56 | 94.26 | 92.67 | 93.16 | 91.77 | **92.18** | 62.57 | 79.64 | 86.41 | 97.09 | 87.04 | 82.55 |
| CIFAR-10 Strong | ViM | 87.75 | 89.62 | 94.76 | 94.50 | 95.15 | 89.49 | 91.88 | 65.54 | 78.63 | 89.56 | 97.97 | 90.50 | 84.44 |
| CIFAR-10 Strong | fDBD * | 87.18 | 89.11 | 90.80 | 87.45 | 90.54 | 91.09 | 89.36 | 70.66 | 82.60 | 93.70 | 92.11 | 91.17 | 86.05 |
| ImageNet Strong | GradNorm | 54.43 | 55.37 | 63.72 | 53.91 | 52.07 | 60.50 | 56.66 | 71.90 | 74.02 | 93.89 | 92.05 | 84.82 | 83.33 |
| ImageNet Strong | NECO | 85.50 | 88.23 | 96.12 | 92.24 | 88.56 | 89.54 | 90.03 | 74.79 | 82.42 | 92.43 | 89.18 | 90.80 | 85.93 |
| ImageNet Strong | ReAct | 85.93 | 88.29 | 92.81 | 89.12 | 89.38 | 90.35 | 89.32 | 73.03 | 81.73 | 96.34 | 92.79 | 91.87 | 87.15 |
| ImageNet Strong | DICE | 77.01 | 79.67 | 90.37 | 90.02 | 81.89 | 74.67 | 82.27 | 70.13 | 76.01 | 92.54 | 92.04 | 88.26 | 83.80 |
| ImageNet Strong | ASH | 74.11 | 76.44 | 83.16 | 73.46 | 77.45 | 79.89 | 77.41 | 72.89 | 83.45 | 97.07 | 96.90 | 93.26 | 88.71 |
| ImageNet Strong | Scale | 80.57 | 83.86 | 93.19 | 86.06 | 83.48 | 88.89 | 86.01 | 77.34 | 85.37 | 98.02 | 96.75 | 93.95 | **90.28** |
| | NCI w/o filter* | 87.93 | 89.66 | 91.50 | 90.81 | 92.18 | 90.74 | 90.47 | 66.81 | 80.20 | 92.67 | 91.87 | 90.51 | 84.41 |
| | NCI | 87.92 | 89.65 | 91.50 | 90.80 | 92.17 | 90.74 | 90.46 | 73.90 | 83.46 | 96.95 | 96.63 | 92.98 | 88.56 |

(a) `NCI` ranks **top-three** in both benchmarks, while baselines show greater variability. ↑ and ↓ denotes better performance. **Bold** marks best, underline 2nd / 3rd. Methods with * are hyperparameter-free. Scores, except for the most recent baselines – `fDBD`, `NECO`, `ASH`, `Scale` – are from OpenOOD Zhang et al. (2023).

| Performance | MSP | NECO | KNN | ViM | ASH | Scale | NCI (ours) |
|---|---|---|---|---|---|---|---|
| CIFAR-10 Latency | 0.53 | 0.70 | 1.95 | 0.70 | 0.53 | 0.53 | 0.54 |
| ImageNet Latency | 6.85 | 9.55 | 10.31 | 9.55 | 7.02 | 7.01 | 6.84 |
| Avg AUROC | 85.69 | 87.98 | 87.38 | 88.16 | 83.06 | 88.15 | **89.51** |

(b) `NCI` improves the **overall performance** while **reducing latency** compared to strong baselines. AUROC averaged across CIFAR-10 and ImageNet benchmarks in Table 1a, with per image latency reported.

from Gaussian $N(0, 1)$, following Sun et al. (2021); Sun & Li (2022). For detailed setups, please see Appendix A. Our method and all baselines are *post-hoc* methods, while all models used are *off-the-shelf* and do *not* require complete Neural Collapse Convergence.

### 4.1 Mitigating Discrepencies across Classification Tasks

We first assess the performance of `NCI` and baselines across CIFAR-10 and ImageNet classification tasks. The two tasks provide an ideal test bed for evaluating versatility, as they drastically differ in input resolution, number of classes, and classification accuracy. We use ResNets from OpenOOD Zhang et al. (2023): ResNet-18 for CIFAR-10 (95.06% accuracy) and ResNet-50 for ImageNet (76.65% accuracy). Based on validation results, we set the filter strength $\alpha$ of the $L1$-norm to $10^{-2}$ for CIFAR-10 experiments and $10^{-3}$ for ImageNet experiments.

Table 2: NCI reduces discrepencies and improves **overall performance** on ImageNet benchmarks across ViT B/16 and Swin v2 classifiers. **Bold** marks best, underline 2nd

| Methods | ImageNet OpenOOD Benchmark (ViT B/16) | | | | | | ImageNet OpenOOD Benchmark (Swin v2) | | | | | |
|---|---|---|---|---|---|---|---|---|---|---|---|---|
| | SSB-hard | NINCO | iNaturalist | Texture | OpenImage-O | AVG | SSB-hard | NINCO | iNaturalist | Texture | OpenImage-O | AVG |
| *Evaluation under FPR95 ↓* | | | | | | | | | | | | |
| KNN | 63.41 | 39.71 | 6.84 | 43.12 | 18.30 | 34.28 | 90.88 | 83.16 | 76.88 | 60.43 | 67.14 | 75.70 |
| ViM | 51.91 | 37.10 | 5.67 | 39.29 | 17.51 | 30.30 | 90.34 | 83.89 | 70.98 | 65.90 | 68.68 | 75.96 |
| ASH | 48.78 | 45.42 | 11.00 | 42.37 | 20.33 | 35.58 | 93.80 | 93.93 | 87.58 | 97.27 | 91.14 | 92.74 |
| Scale | 45.07 | 32.04 | 5.49 | 40.59 | 13.15 | **27.27** | 90.74 | 75.72 | 48.73 | 95.10 | 64.55 | 75.97 |
| NCI w/o filter | 50.94 | 30.68 | 5.93 | 46.61 | 14.92 | 29.81 | 86.77 | 73.11 | 47.98 | 75.30 | 59.30 | 69.67 |
| NCI | 46.73 | 33.79 | 6.08 | 42.09 | 14.79 | 28.79 | 85.58 | 72.06 | 45.25 | 71.53 | 54.72 | **65.83** |
| *Evaluation under AUROC ↑* | | | | | | | | | | | | |
| KNN | 81.48 | 90.00 | 98.67 | 96.23 | 96.23 | 91.44 | 62.50 | 69.74 | 78.35 | 85.19 | 67.14 | 75.88 |
| ViM | 87.39 | 92.56 | 98.98 | 90.80 | 96.82 | 93.31 | 60.99 | 72.30 | 83.36 | 79.61 | 82.52 | 75.76 |
| ASH | 90.60 | 90.88 | 98.04 | 95.97 | 95.97 | 93.14 | 58.87 | 58.28 | 58.18 | 46.18 | 61.32 | 56.57 |
| Scale | 89.67 | 93.23 | 98.96 | 97.20 | 97.20 | **94.09** | 62.48 | 78.97 | 88.88 | 67.08 | 86.14 | 76.71 |
| NCI w/o filter | 87.16 | 93.15 | 98.87 | 96.80 | 96.80 | 93.26 | 64.53 | 76.73 | 88.07 | 79.63 | 85.39 | 78.87 |
| NCI | 88.86 | 92.88 | 98.79 | 96.83 | 96.83 | 93.64 | 67.53 | 78.99 | 89.68 | 81.43 | 87.42 | **80.97** |

(a) NCI boosts Swin v2 while maintaining ViT effectiveness compared to baselines, even without filtering.

| Performance | KNN | ViM | ASH | Scale | NCI (ours) |
|---|---|---|---|---|---|
| Avg AUROC | 83.66 | 84.84 | 74.86 | 85.40 | **87.31** |

(b) NCI improves the **overall performance**. AUROC averaged across two architectures in Table 2a

**Datasets** For CIFAR-10 experiments, We follow the OpenOOD split of ID test set and evaluate on the OpenOOD benchmarks, including CIFAR-100 Krizhevsky et al. (2009), Tiny ImageNet Le & Yang (2015), MNIST Deng (2012), SVHN Netzer et al. (2011), Texture (Cimpoi et al., 2014), and Places365 (Zhou et al., 2017). For ImageNet experiments, we follow the OpenOOD split of ID test set and evaluate on the OpenOOD benchmarks, including SSB-hard Vaze et al. (2021), NINCO Bitterwolf et al. (2023), iNaturalist (Van Horn et al., 2018), Texture (Cimpoi et al., 2014), and OpenImage-O Wang et al. (2022).

**Baselines** In Table 1a, we compare our method with *thirteen* baselines. Some baselines focus more on the CIFAR-10 Benchmark while others focus more focused on the Imagenet Benchmark. Therefore, we categorize the baselines, besides the vanilla confidence-based MSP (Hendrycks & Gimpel, 2016), into two groups: the "CIFAR-10 Strong" baselines, including ODIN (Liang et al., 2018), Energy (Liu et al., 2020), Mahalanobis (Lee et al., 2018), KNN(Sun et al., 2022), ViM (Wang et al., 2022), and fDBD Liu & Qin (2023); the "ImageNet Strong" baselines, including GradNorm (Huang et al., 2021), NECO Ammar et al. (2023), React (Sun et al., 2021), Dice (Sun & Li, 2022), ASH Djurisic et al. (2022), Scale Xu et al. (2023). See Appendix C for details of the baselines.

**Performance** Table 1a shows that NCI consistently ranks *top-three* across benchmarks, whereas baselines exhibit greater variability. To assess overall performance, we averaged AUROC across benchmarks, which are of a similar range. Table 1b highlights that NCI improves *overall* performance compared to strong baselines on individual benchmarks. Further, NCI is as efficient as MSP, as shown in Table 1b[1], which enhances efficiency compared to strong baselines. This aligns with the analysis in Section 3 and Appendix C. We highlight the following pairs of comparison:

- NCI **v.s.** NCI **w/o filter:** On the CIFAR-10 classifier, strong ID clustering allows our method to rank top-3 without filtering. Conversely, on the ImageNet ResNet-50, weaker ID clustering (see Appendix E) makes norm-based filtering crucial for reducing generalization discrepancy. Complete Neural Collapse occurs on neither model while NCI remains effective.

- NCI **v.s.** KNN: Compared to KNN, NCI significantly reduces the latency (Table 1b). Notably, without filtering, our hyperparameter-free score outperforms KNN with tuned parameters on most benchmarks (Table 1a, Table 2a & Table 8), highlighting the benefit of using class-specific information.

- NCI **v.s.** ASH / Scale: Compared to both, NCI delivers competitive performance on ImageNet and *significantly* improves CIFAR-10, enhancing *overall* performance ( Table 1b). Also, ASH and Scale introduce in a small delay on the ImageNet benchmark due to activation sorting, with larger activation dimensions likely widening the latency gap on larger models.

---

[1]Running time of KNN on ImageNet are copied from Table 4 in Sun et al. (2022).

Table 3: NCI improves the **overall performance**, averaged across Table 1a, Table 2a & Table 8.

| Performance | KNN | ViM | ASH | Scale | NCI (ours) |
|---|---|---|---|---|---|
| Avg AUROC Across **All** Benchmarks | 86.06 | 85.96 | 81.24 | 86.8 | **88.57** |

- NCI **v.s.** NECO: NECO (Ammar et al., 2023) is motivated by Neural Collapse. Like NCI with filtering, NECO uses max-logit and incorporates distance to the origin. However, NECO exclusively analyzes features, requiring expensive matrix multiplication and leading to higher inference latency (Table 1b). Conversely, NCI explores the interplay between features *and* the classification head, integrating class-specific information to improve both efficiency and effectiveness.

## 4.2 MITIGATING DISCREPANCIES ACROSS ARCHITECTURES

Next, we study two transformer-based models: ViT B/16 Dosovitskiy et al. (2020) and Swin-v2 Liu et al. (2022), both finetuned on ImageNet, achieving an accuracy of 81.14% and 82.94% respectively. We follow the setup of the OpenOOD ImageNet Benchmark in Section 4.1. Based on validation results, we set the filter strength $\alpha$ of the $L1$ norm to $10^{-3}$ for both classifiers. In Table 2, we observe strong baselines suffer on Swin v2, echoing the observations in Ammar et al. (2023). Conversely, our NCI, even without filtering, improves baseline performance on Swin v2. Filtering further enhances the performance, leading to improved *overall* performance (Table 2b).

We further aggregate in Table 2 with experiments on ResNet (Table 1) and DenseNet (Table 8) and report the average AUROC in Table 3. NCI significantly boosts the overall performance.

## 4.3 ABLATION ON THE FILTERING EFFECT

In Table 4, we assess different orders of $p$-norm as the filtering term, compared to the $L1$ norm used so far. To ensure a fair comparison, we report the best performance from the filter strengths $\{10^{-4}, 10^{-3}, 10^{-2}, 10^{-1}\}$. The rest of the setup follows the ImageNet benchmarks in Section 4.1. As shown in Table 4, filtering with $L1$ norm achieves the best performance across OOD datasets, aligning with prior observations Huang et al. (2021); Park et al. (2023). Meanwhile, we observe that in rare scenarios, e.g., a ResNet-18 on CIFAR-10, the L1 norm cannot effectively characterize OOD's proximity to the origin, leading to no extra performance gain compared to simply thresholding on pScore. In these cases, our algorithm benefits from its ability to automatically select a low filter strength based on validation results, effectively disregarding the filtering term.

Table 4: Ablation on filtering norm on ImageNet OpenOOD Benchmark with ResNet-50 backbone. AUROC score is reported (higher is better). **Bold** denotes the best result. Filtering with L1 norm outperforms alternative choice of norms across OOD datasets.

| | SSB-hard | NINCO | iNaturalist | Texture | OpenImage-O |
|---|---|---|---|---|---|
| Filtering w/ Linf | 66.81 | 80.20 | 92.66 | 91.87 | 90.51 |
| Filtering w/ L2 | 69.12 | 81.44 | 93.96 | 92.77 | 91.73 |
| Filtering w/ L1 | **73.90** | **83.46** | **96.95** | **96.63** | **92.98** |

We also test the sensitivity of NCI to filtering strength $\alpha$ in Table add. As shown on the ImageNet ResNet50 benchmark, performance remains stable for $\alpha$ values within the same scale. Given this insensitivity, we select hyperparameters from four scales $\{10^{-4}, 10^{-3}, 10^{-2}, 10^{-1}\}$ without extensive finetuning in this work.

Table 5: Sensitivity of NCI to filtering strength. Average AUROC on ImageNet ResNet-50 Benchmark reported. Performance remains stable within the same scale.

| Filtering Strength $\alpha$ | 0.6 ×10-3 | 0.8 × 10-3 | 1.0 ×10-3 | 1.2 × 10-3 | 1.4 × 10-3 |
|---|---|---|---|---|---|
| Avg AUROC | 88.27 | 88.55 | 88.59 | 88.50 | 88.23 |

We further apply L1-norm based filtering to KNN to see if this perspective can mitigate the discrepancy of clustering-based methods in general. In Table 6 [2], we report the the best performance of KNN from filter strengths $\{10^{-4}, 10^{-3}, 10^{-2}, 10^{-1}\}$. We observe a significant performance gain from adding the filter, which further validates our understanding of ID clustering landscape from Neural Collapse. Note that our method outperforms the standalone $L1$ norm as well as KNN, before and after filtering.

Table 6: Effectiveness of our filtering scheme on KNN. Performance gain validates our understanding of ID clustering landscape. NCI outperforms KNN and standalone $L1$ norm. AUROC reported (higher is better). **Bold** highlights the best result.

|  | SSB-hard | NINCO | iNaturalist | Texture | OpenImage-O | AVG |
|---|---|---|---|---|---|---|
| L1 | 68.80 | 68.28 | 90.86 | 88.16 | 78.47 | 78.91 |
| KNN | 62.57 | 79.64 | 86.41 | 96.49 | 87.04 | 82.43 |
| KNN + L1 | 64.29 | 81.76 | 92.76 | 97.85 | 90.17 | 86.37 |
| NCI w/o L1 | 66.81 | 80.20 | 92.67 | 91.87 | 90.51 | 84.41 |
| NCI | 73.90 | 83.46 | 96.95 | 96.63 | 92.98 | **88.56** |

## 5 RELATED WORK

**OOD Detection** Extensive research has focused on OOD detection algorithms. One line of work is post-hoc and builds upon pre-trained models. For example, Hendrycks et al. (2019); Liang et al. (2018); Liu et al. (2020); Sun et al. (2021); Sun & Li (2022); Liu & Qin (2023); Xu et al. (2024) design OOD score over the output space of a classifier. Meanwhile, Lee et al. (2018) and Sun et al. (2022) measure OOD-ness from the perspective of ID clustering in *feature* space. Our work extends the observation that ID features tend to cluster from the perspective of Neural Collapse. While existing work is more focused are certain classification tasks than others, our proposed OOD detector is tested to be highly versatile.

Others (Sharifi et al., 2024; Patil et al., 2024; Zhu et al., 2024) explore the regularization of OOD detection in training. For example, DeVries & Taylor (2018); Hsu et al. (2020) propose OOD-specific architecture whereas Huang & Li (2021); Wei et al. (2022) design OOD-specific training loss. In particular, Tack et al. (2020) brings attention to representation learning for OOD detection and proposes an OOD-specific contrastive learning scheme. Our work does not belong to this school of thought and is not restricted to specific training schemes or architecture.

**Neural Collapse** Neural Collapse was first observed in Papyan et al. (2020). During Neural Collapse, the penultimate layer features collapse to class means, the class means and the classifier collapses to a simplex equiangular tight framework, and the classifier simplifies to adopt the nearest class-mean decision rule. Further work provides theoretical justification for the emergence of Neural Collapse (Han et al., 2021; Mixon et al., 2020; Zhou et al., 2022; Zhu et al., 2021). In addition, Zhu et al. (2021) derives an efficient training algorithm drawing inspiration from Neural Collapse. Our concurrent work Ammar et al. (2023) also leverages insights from Neural Collapse for OOD detection. However, they tackle from the subspace perspective and largely overlook class-specific information revealed by Neural Collapse, which is essential for our work.

## 6 CONCLUSION

This work leverages insights from Neural Collapse to propose a novel OOD detector. Specifically, we study the phenomenon that ID features tend to form clusters whereas OOD features reside far away. Inspired by the trend of Neural Collapse prevalent on practical models, we hypothesize and validate that ID features tend to cluster near weight vectors. We also explain why ID features tend to reside further from the origin and complement our method from this perspective. Experiments show the effectiveness of our method on practical models without requiring the complete convergence of Neural Collapse. Further, our method improves the overall performance with minimal latency across diverse benchmarks. We hope our work can inspire future work to explore the interplay between features and weight vectors for OOD detection and other research problems such as calibration and adversarial robustness.

---

[2]Note that we report our run of KNN here to ensure a fair evaluation of the filtering effect. Our results are very similar to the OpenOOD results reported in Table 1a with only marginal differences.

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
