# A    IMPLEMENTATION DETAILS

## A.1    CIFAR-10

**ResNet-18** For visualization in Fig. 1*Left, Middle*, we use a CIFAR-10 classifier of ResNet-18 backbone trained with cross-entropy loss. The classifier is trained for 100 epochs, with the initial learning rate 0.1 decaying to 0.01, 0.001, and 0.0001 at epochs 50, 75, and 90 respectively. For experiments in Table 1a, we use the pre-trained model provided by the OpenOOD benchmark. And we refer readers to Zhang et al. (2023) for their training recipe.

**DenseNet-101** For experiments on CIFAR-10 Benchmark presented in Table 8, we evaluate a CIFAR-10 classifier of DenseNet-101 backbone. The classifier is trained following the setups in Huang et al. (2017) with depth $L = 100$ and growth rate $k = 12$.

## A.2    CIFAR-100

**DenseNet-101** For experiments on the CIFAR-100 Benchmark presented in Table 8, we evaluate a CIFAR-100 classifier of the DenseNet-101 backbone. The classifier is trained following the setups in Huang et al. (2017) with depth $L = 100$ and growth rate $k = 12$.

## A.3    IMAGENET

**ResNet-50** For evaluation on ImageNet Benchmark in Table 1a, we use the default ResNet-50 model trained with cross-entropy loss provided by Pytorch. Training recipe can be found at https://pytorch.org/blog/how-to-train-state-of-the-art-models-using-torchvision-latest-primitives/

**ViT B/16** In Table 2, we use the PyTorch implementation and pre-trained checkpoint of ViT B/16, available https://github.com/lukemelas/PyTorch-Pretrained-ViT/tree/master.

**Swin v2** In Table 2, we use the `timm` Wightman (2019) implementation of Swin v2 as well as their pre-trained checkpoint 'swinv2_base_window8_256'.

# B    ALTERNATIVES PROXIMITY METRICS

In this section, we validate that under alternative similarity metrics, ID features also reside closer to weight vectors and empirically compare the metrics. In addition to our proposed pScore, we consider two standard similarity metrics, cosine similarity and Euclidean distance. For cosine similarity, we evaluate

$$\text{cosScore} = \frac{(\boldsymbol{h} - \boldsymbol{\mu}_G) \cdot \boldsymbol{w}_c}{\|\boldsymbol{h} - \boldsymbol{\mu}_G\|_2 \|\boldsymbol{w}_c\|_2}. \tag{7}$$

As for Euclidean distance, we first estimate the scaling factor in Theorem 3.1 by $\tilde{\lambda}_c = \frac{\|\boldsymbol{\mu}_c - \boldsymbol{\mu}_G\|_2}{\|\boldsymbol{w}_c\|_2}$.

Based on the estimation, we measure the distance between the centered feature $\boldsymbol{h} - \boldsymbol{\mu}_G$ and the scaled weight vector corresponding to the predicted class $c$ as

$$\text{distScore} = -\|(\boldsymbol{h} - \boldsymbol{\mu}_G) - \tilde{\lambda}_c \boldsymbol{w}_c\|_2. \tag{8}$$

Same as pScore, the larger cosScore or distScore is, the closer the feature is to the weight vector.

We evaluate in Table 7 OOD detection performance using standalone pScore, cosScore, and distScore as scoring function respectively. The experiments are evaluated with AUROC under the same ImageNet setup as in Section 4.1. We observe in Table 7, that across OOD datasets, all three scores achieve an AUROC score $> 50$, indicating that ID features reside closer to weight vectors compared to OOD under either metric.

Furthermore, we observe that pScore outperforms both cosScore and distScore. Comparing the performance of pScore and cosScore, the superior performance of pScore implies that ID features corresponding to the classes with larger $\boldsymbol{w}_c$ are less compact. This is in line with the decision rule of the classifier that classes with larger $\boldsymbol{w}_c$ have larger decision regions. As for comparison against Euclidean distance based distScore, pScore eliminates the need to estimate the scaling factor, which can be error-prone before convergence, potentially leading to performance degradation.

Table 7: Ablation on proximity scores. AUROC score is reported (higher is better). ID features are closer to weight vectors than OOD features (AUROC > 50) under all metrics. Across OOD datasets, our proposed `pScore` can better separate ID an OOD features than `distScore` and `cosScore`.

|  | SSB-hard | NINCO | iNaturalist | Texture | OpenImage-O |
|---|---|---|---|---|---|
| distScore | 54.69 | 70.20 | 85.78 | 87.07 | 78.46 |
| cosScore | 65.82 | 79.92 | 90.43 | 91.36 | 89.00 |
| pScore | 66.81 | 80.20 | 92.67 | 91.87 | 90.51 |

## C  BASELINE METHODS

We provide an overview of our baseline methods in this session. We follow our notation in Section 3. In the following, a lower detection score indicates OOD-ness.

**MSP** Hendrycks & Gimpel (2016) proposes to detect OOD based on the maximum softmax probability. Given the penultimate feature $h$ for a given test sample $x$, the detection score of MSP can be represented as:

$$\frac{\exp\left(\boldsymbol{w}_c^T \boldsymbol{h} + b_c\right)}{\sum_{c' \in \mathcal{C}} \exp\left(\boldsymbol{w}_{c'}^T \boldsymbol{h} + b_{c'}\right)}, \tag{9}$$

where $c$ is the predicted class for $x$.

**ODIN** Liang et al. (2018) proposes to amplify ID the OOD separation on top of MSP through temperature scaling and adversarial perturbation. Given a sample $x$, ODIN constructs a noisy sample $x'$ from $x$. Denote the penultimate feature of the noisy sample $x'$ as $h'$, ODIN assigns OOD score following:

$$\frac{\exp\left((\boldsymbol{w}_c^T \boldsymbol{h'} + b_c)/T\right)}{\sum_{c' \in \mathcal{C}} \exp\left((\boldsymbol{w}_c'^T \boldsymbol{h'} + b_{c'})/T\right)}, \tag{10}$$

where $c$ is the predicted class for the perturbed sample and $T$ is the temperature. In our implementation, we set the noise magnitude as 0.0014 and the temperature as 1000.

**Energy** Liu et al. (2020) designs an energy-based score function over the logit output. Given a test sample $x$ as well as its penultimate layer feature $h$, the energy based detection score can be represented as:

$$-\log \sum_{c' \in \mathcal{C}} \exp\left(\boldsymbol{w}_{c'}^T \boldsymbol{h} + b_{c'}\right). \tag{11}$$

**ReAct** Sun et al. (2021) builds upon the energy score proposed in Liu et al. (2020) and regularizes the score by truncating the penultimate layer estimation. We set the truncation threshold at 90 percentile in our experiments.

**Dice** Sun & Li (2022) builds upon the energy score proposed in Liu et al. (2020). Leveraging the observation that units and weights are used sparsely in ID inference, Sun & Li (2022) proposes to select and compute the energy score over a selected subset of weights based on their importance. We set a threshold at 90 percentile for CIFAR experiments and 70 percentile for ImageNet experiments following Sun & Li (2022).

**ASH** Djurisic et al. (2022) builds upon the energy score proposed in Liu et al. (2020). Prior to the Energy score, ASH sorts each feature to find the top-k elements, scales up the top-k elements, and sets the rest to zero. We note that in addition to the cost of Energy, ASH introduces a sorting cost of $O(P \log k)$, where $P$ is the penultimate layer dimension.

**Scale** Xu et al. (2023) builds upon the energy score proposed in Liu et al. (2020). Prior to the Energy score, Scale sorts each feature to find the top-k elements and based on the statistics, scales

all elements in the feature. We note that in addition to the cost of Energy, Scale also introduces a sorting cost of $O(P \log k)$, where $P$ is the penultimate layer dimension.

**Mahalanobis** On the feature space, Lee et al. (2018) models the ID feature distribution as multivariate Gaussian and designs a Mahalanobis distance-based score:

$$\max_c -(\boldsymbol{e_x} - \hat{\boldsymbol{\mu}}_c)^T \hat{\Sigma}^{-1} (\boldsymbol{e_x} - \hat{\boldsymbol{\mu}}_c), \tag{12}$$

where $\boldsymbol{e_x}$ is the feature embedding of $\boldsymbol{x}$ in a specific layer, $\hat{\mu}_c$ is the feature mean for class $c$ estimated on the training set, and $\hat{\Sigma}$ is the covariance matrix estimated over all classes on the training set.

On top of the basic score, Lee et al. (2018) also proposes two techniques to enhance the OOD detection performance. The first is to inject noise into samples. The second is to learn a logistic regressor to combine scores across layers. We tune the noise magnitude and learn the logistic regressor on an adversarial constructed OOD dataset. The selected noise magnitude is 0.005 in both our ResNet and DenseNet experiments.

**KNN** Chen et al. (2020) proposes to detect OOD based on the k-th nearest neighbor distance between the normalized embedding of the test sample $\boldsymbol{z_x}/|\boldsymbol{z_x}|$ and the normalized training embeddings on the penultimate space. Chen et al. (2020) also observes that contrastive learning helps in improving OOD detection effectiveness.

**GradNorm** Huang et al. (2021) extracts information from the gradient space to detect OOD samples. Specifically, Huang et al. (2021) defines the OOD score function as the L1 norm of the gradient of the weight matrix with respect to the KL divergence between the softmax prediction for $\boldsymbol{x}$ and the uniform distribution.

$$\|\frac{\partial D_{KL}(\boldsymbol{u}\|softmax f(\boldsymbol{x}))}{\partial \boldsymbol{W}}\|_1. \tag{13}$$

**ViM** Wang et al. (2022) proposes to integrate class-specific information into feature space information by adding energy score to the feature norm in the residual space of the training feature matrix. The detection score is designed to be:

$$\alpha \sqrt{\boldsymbol{h}^T \boldsymbol{R} \boldsymbol{R} \boldsymbol{h}}, \tag{14}$$

where $\boldsymbol{R} \in R^{P \times (P-D)}$ correspond to the residual after subtracting the $D-$dimensional principle space. In the preparation stage, ViM requires evaluating the residual/null space from the training data, which is computationally expensive given the data volume. During inference, large matrix multiplication is required, resulting in a computational complexity of $O((P-D)^2)$.

**NECO** is inspired by the ETF structure of Neural Collapse to utilize feature subspace for OOD detection. The detection score is designed to be

$$\text{MaxLogit} \times \frac{\sqrt{\boldsymbol{h}^T \boldsymbol{P} \boldsymbol{P} \boldsymbol{h}}}{\sqrt{\boldsymbol{h}^T \boldsymbol{h}}}, \tag{15}$$

where $\boldsymbol{P} \in R^{P \times d}$ correspond to the $d-$dimensional principle space. In the preparation stage, NECO requires evaluating the residual/null space from the training data, which is computationally expensive given the data volume. During inference, large matrix multiplication is required, resulting in a computational complexity of $O((d)^2 + P)$.

**fDBD** Liu & Qin (2023) proposes to detect OOD based on estimated feature distance to decision boundaries of class $c \in \mathcal{C}$ besides its predicted class $f(\boldsymbol{x})$:

$$\tilde{D}_f(\boldsymbol{h}, c) = \frac{|(\boldsymbol{w}_{f(\boldsymbol{x})} - \boldsymbol{w}_c)^T \boldsymbol{h} + (b_{f(\boldsymbol{x})} - b_c)|}{\|\boldsymbol{w}_{f(\boldsymbol{x})} - \boldsymbol{w}_c\|_2}, \tag{16}$$

The detection score is designed as

$$\frac{1}{|\mathcal{C}| - 1} \sum_{c \in \mathcal{C}, \, c \neq f(\boldsymbol{x})} \frac{\tilde{D}_f(\boldsymbol{h}, c)}{\|\boldsymbol{h} - \boldsymbol{\mu}_{train}\|_2}. \tag{17}$$

fDBD has time complexity $O(|\mathcal{C}| + P)$, where $|\mathcal{C}|$ is the number of training classes and $P$ is the penultimate layer dimension.

Table 8: **Our OOD detectors achieves high AUROC and low FPR95 across CIFAR-10 and CIFAR-100 OOD benchmark on DenseNet.** ↑ indicates that larger values are better and vice versa. **Bold** highlight the best results and underline denotes the 2nd and 3rd best results. We note that for DenseNet CIFAR-10 and CIFAR-100 classifiers, the discrepancy among existing methods is not as severe as in the examples presented in the main paper. Nevertheless, our NCI achieves state-of-the-art performance or improves upon existing methods, enhancing overall performance on average.

| Methods | CIFAR-10 OOD Benchmark | | | | | | | CIFAR-100 OOD Benchmark | | | | | | |
|---|---|---|---|---|---|---|---|---|---|---|---|---|---|---|
| | CIFAR-100 | TIN | MNIST | SVHN | Texture | Place365 | AVG | CIFAR-10 | TIN | MNIST | SVHN | Texture | Place365 | Avg |
| *Evaluation under FPR95 ↓* | | | | | | | | | | | | | | |
| MSP | 36.46 | 31.51 | 20.79 | 19.02 | 39.17 | 32.69 | 29.04 | 65.62 | 59.33 | 61.30 | 74.09 | 78.97 | 62.53 | 66.97 |
| ODIN | 41.11 | 32.89 | 11.19 | 27.03 | 49.98 | 30.61 | 32.13 | 72.72 | 56.67 | 60.23 | 52.44 | 83.88 | 57.58 | 63.92 |
| Energy | 38.73 | 29.17 | 9.46 | 17.41 | 58.06 | 30.26 | 30.51 | 75.30 | 54.82 | 54.33 | 49.64 | 93.14 | 59.59 | 94.47 |
| MDS | 88.91 | 89.17 | 70.42 | 49.48 | 68.41 | 90.72 | 76.27 | 90.04 | 87.80 | 54.20 | 80.69 | 62.61 | 88.71 | 77.34 |
| KNN | 40.42 | 33.97 | 12.97 | 4.71 | 19.97 | 37.08 | 24.84 | 84.20 | 66.64 | 19.46 | 22.59 | 36.88 | 74.86 | **50.76** |
| ViM | 42.74 | 35.67 | 14.16 | 19.72 | 24.81 | 36.53 | 28.94 | 76.78 | 59.07 | 67.34 | 54.06 | 34.74 | 63.60 | 59.27 |
| fDBD | 38.87 | 31.29 | 10.32 | 6.70 | 18.32 | 31.30 | 22.80 | 68.17 | 53.08 | 43.03 | 45.80 | 35.66 | 62.90 | 51.44 |
| GradNorm | 72.67 | 55.37 | 8.57 | 21.94 | 86.36 | 63.97 | 51.48 | 94.07 | 84.61 | 41.99 | 36.54 | 97.98 | 81.32 | 72.75 |
| NECO | 38.51 | 29.12 | 9.68 | 16.91 | 56.29 | 29.94 | 33.82 | 75.16 | 54.63 | 54.18 | 49.73 | 92.07 | 59.34 | 63.91 |
| ReAct | 35.99 | 27.34 | 10.78 | 15.63 | 32.87 | 27.12 | 24.96 | 72.48 | 54.08 | 47.47 | 52.76 | 71.38 | 60.28 | 59.74 |
| DICE | 46.47 | 33.12 | 5.23 | 17.52 | 65.39 | 36.36 | 34.02 | 88.20 | 67.38 | 57.39 | 37.62 | 91.93 | 61.91 | 67.40 |
| ASH | 46.16 | 32.67 | 12.44 | 12.61 | 42.76 | 30.71 | 29.56 | 84.20 | 66.14 | 44.44 | 33.29 | 69.00 | 69.96 | 61.17 |
| Scale | 38.12 | 26.82 | 7.51 | 9.41 | 40.66 | 28.63 | 25.19 | 77.97 | 54.12 | 48.74 | 38.84 | 81.73 | 58.93 | 60.05 |
| NCI (Ours) | 36.08 | 29.50 | 8.44 | 5.67 | 16.22 | 30.83 | **21.12** | 84.99 | 57.33 | 29.71 | 25.99 | 50.16 | 64.40 | 52.10 |
| *Evaluation under AUROC ↑* | | | | | | | | | | | | | | |
| MSP | 87.97 | 89.52 | 92.79 | 93.30 | 87.29 | 89.25 | 90.02 | 74.11 | 76.74 | 74.42 | 68.40 | 69.99 | 75.14 | 73.14 |
| ODIN | 88.94 | 91.31 | 97.28 | 93.28 | 87.67 | 92.17 | 91.78 | 73.20 | 80.86 | 77.30 | 76.55 | 74.24 | 81.01 | 77.20 |
| Energy | 89.38 | 92.37 | 97.54 | 94.74 | 85.49 | 92.52 | 92.00 | 73.50 | 81.71 | 78.66 | 78.38 | 69.63 | 79.60 | 76.92 |
| MDS | 60.33 | 56.43 | 63.17 | 90.15 | 88.42 | 56.63 | 69.19 | 50.41 | 57.26 | 74.78 | 70.14 | 88.67 | 56.80 | 66.34 |
| KNN | 88.75 | 90.78 | 96.61 | 99.13 | 96.14 | 90.42 | 93.63 | 60.59 | 73.97 | 93.89 | 94.24 | 92.88 | 68.18 | 80.63 |
| ViM | 87.71 | 89.64 | 95.82 | 95.20 | 95.16 | 89.50 | 92.17 | 67.93 | 78.37 | 70.73 | 78.70 | 93.12 | 76.78 | 77.60 |
| fDBD | 89.98 | 92.04 | 97.52 | 98.34 | 95.58 | 92.17 | 94.27 | 75.83 | 82.37 | 84.46 | 85.05 | 90.26 | 77.79 | 82.63 |
| GradNorm | 78.47 | 85.19 | 97.91 | 95.85 | 83.14 | 83.18 | 87.29 | 51.75 | 64.64 | 86.41 | 89.63 | 73.16 | 66.61 | 72.03 |
| NECO | 89.43 | 92.38 | 97.44 | 94.93 | 85.87 | 92.53 | 92.10 | 73.77 | 81.76 | 78.83 | 78.58 | 70.40 | 79.62 | 77.30 |
| ReAct | 90.06 | 92.67 | 97.17 | 94.98 | 90.77 | 93.03 | 93.11 | 74.38 | 81.86 | 81.65 | 79.02 | 76.47 | 78.82 | 78.70 |
| DICE | 86.71 | 91.17 | 98.84 | 96.23 | 86.59 | 91.01 | 91.76 | 59.87 | 76.21 | 80.45 | 89.39 | 77.20 | 79.32 | 77.07 |
| ASH | 87.55 | 91.29 | 96.84 | 96.95 | 90.60 | 91.76 | 92.50 | 66.25 | 76.46 | 86.38 | 89.02 | 83.63 | 72.78 | 79.08 |
| Scale | 89.77 | 93.04 | 98.04 | 97.45 | 90.60 | 92.84 | 93.62 | 73.11 | 81.98 | 82.14 | 85.91 | 77.53 | 79.77 | 80.08 |
| NCI (Ours) | 90.31 | 92.29 | 97.93 | 98.67 | 95.87 | 91.86 | **94.49** | 69.84 | 80.75 | 91.42 | 92.12 | 88.46 | 76.99 | **83.26** |

## D  EVALUATION ON DENSENET

In addition to evaluation on ResNet and transformer-based model in Section 4, we report the performance of our NCI along with the baselines under AUROC and FPR95 across OpenOOD benchmarks in Table 8.

## E  THE PREVALENCE OF NEURAL COLLAPSE ACROSS CANONICAL CLASSIFICATION TASKS

The phenomenon of Neural Collapse, as established in the seminal work by Papyan et al. Papyan et al. (2020) and corroborated by subsequent studies Han et al. (2021); Mixon et al. (2020); Zhou et al. (2022); Zhu et al. (2021), widely exists across canonical classification datasets and model architectures. The prevalent occurrence of Neural Collapse forms a robust foundation for the design of our versatile OOD detectors. To this end, we review the empirical evidence of Neural Collapse across different datasets and model architectures in Figure 3, Figure 4, Figure 5, Figure 6, and Figure 7. Comparing CIFAR-10 and ImageNet behaviors with ResNet backbone in Figure 7, we

note that the clustering of CIFAR-10 is more prominent than Imagenet, as indicated by a higher ratio of between-class variance to within-class covariance. Note that the figures and captions are sourced from Papyan et al. (2020). The definition and notation follow Section 3.

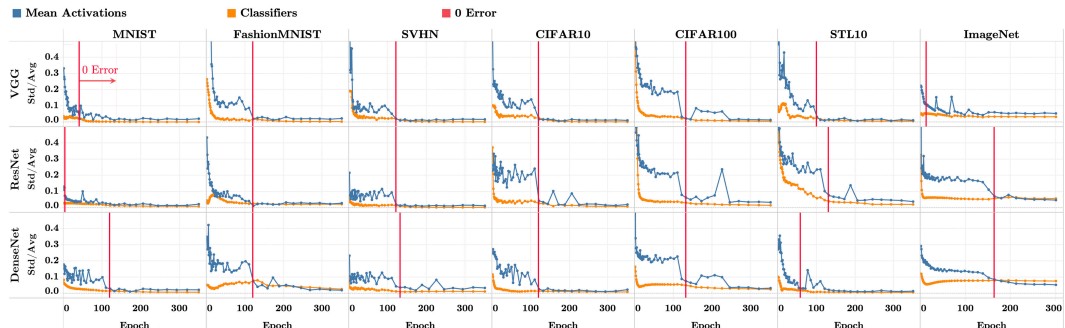

Figure 3: (ref. Figure 2 in Papyan et al. (2020)) **Train class means become equinorm.** In each array cell, the vertical axis shows the coefficient of variation of the centered class-mean norms as well as the network classifiers norms. In particular, the blue lines show $\text{Std}_c(\|\boldsymbol{\mu}_c - \boldsymbol{\mu}_G\|_2)/\text{Avg}(\|\boldsymbol{\mu} - \boldsymbol{\mu}_G\|_2)$ where $\{\boldsymbol{\mu}_c\}$ are the class means of the last-layer activations of the training data and $\boldsymbol{\mu}_G$ is the corresponding train global mean; the orange lines show $\text{Std}_c(\|\boldsymbol{w}_c\|_2)/\text{Avg}(\|\boldsymbol{w}_c\|_2)$ where $\{\boldsymbol{w}_c\}$ is the last-layer classifier of the $c$ th class. As training progresses, the coefficients of variation of both class means and classifiers decrease.

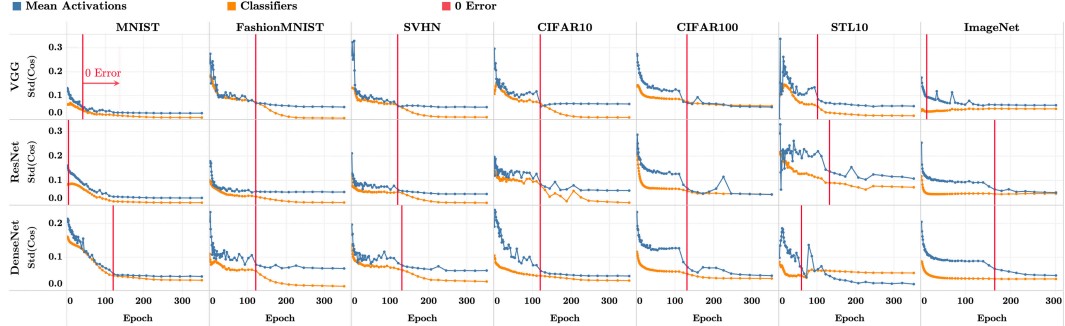

Figure 4: (ref. Figure 3 in Papyan et al. (2020)) **Classifiers and train class means approach equiangularity.** In each array cell, the vertical axis shows the SD of the cosines between pairs of centered class means and classifiers across all distinct pairs of classes $c$ and $c'$. Mathematically, denote $\cos_\mu(c, c') = < \boldsymbol{\mu}_c - \boldsymbol{\mu}_G, \boldsymbol{\mu}'_c - \boldsymbol{\mu}_G > /\|\boldsymbol{\mu}_c - \boldsymbol{\mu}_G\|_2\|\boldsymbol{\mu}'_c - \boldsymbol{\mu}_G\|_2$ and $\cos_w(c, c') = < \boldsymbol{w}_c, \boldsymbol{w}'_c > /\|\boldsymbol{w}_c\|_2\|\boldsymbol{w}'_c\|_2$, where $\{\boldsymbol{w}_c\}_{c=1}^C, \{\boldsymbol{\mu}_c\}_{c=1}^C$, and $\boldsymbol{\mu}_G$ are as in Figure 3. We measure $\text{Std}_{c,c'}(\cos_\mu(c, c'))$ (orange) and $\text{Std}_{c,c'}(\cos_w(c, c'))$. As training progresses, the SDs of the cosines approach zero, indicating equiangularity.

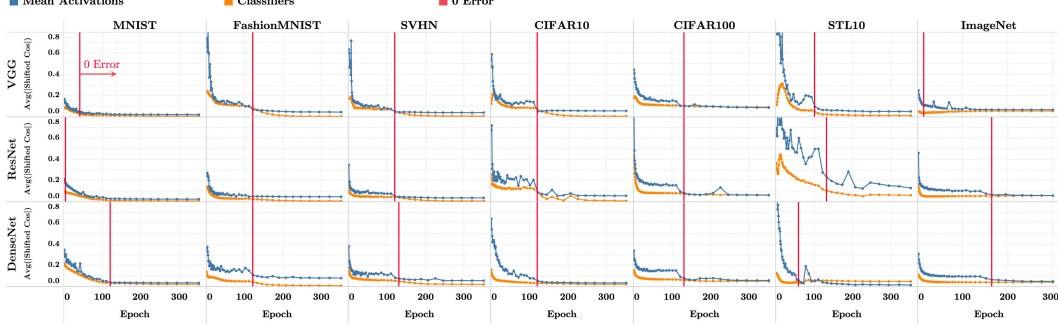

Figure 5: (ref. Figure 4 in Papyan et al. (2020)) **Classifiers and train class means approach maximal-angle equiangularity.** We plot in the vertical axis of each cell the quantities $\text{Avg}_{c,c'}|\cos_\mu(c, c') + 1/(C - 1)|$ (blue) and $\text{Avg}_{c,c'}|\cos_w(c, c') + 1/(C - 1)|$ (orange), where $\cos_\mu(c, c')$ and $\cos_w(c, c')$ are as in Figure 4. As training progresses, the convergence of these values to zero implies that all cosines converge to $-1/(C - 1)$. This corresponds to the maximum separation possible for globally centered, equiangular vectors.

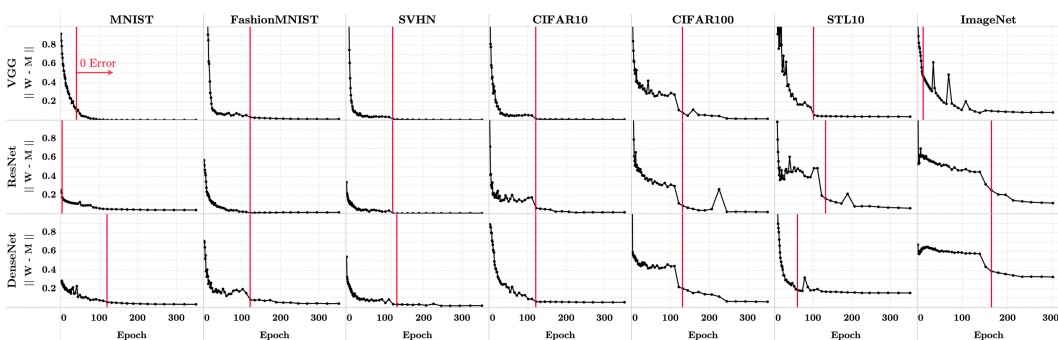

Figure 6: (ref. Figure 5 in Papyan et al. (2020)) **Classifier converges to train class means.** The formatting and technical details are as described in Section 3. In the vertical axis of each cell, we measure the distance between the classifiers and the centered class means, both rescaled to unit norm. Mathematically, denote $\tilde{M} = M/\|M\|_F$ where $M = [\mu_c - \mu_G, c = 1, ...., C] \in \mathbb{R}^{P \times C}$ is the matrix whose columns consist of the centered train class means; denote $\tilde{W} = W/\|W\|_F$ where $W \in \mathbb{R}^{C \times P}$ is the last-layer classifier of the network. We plot the quantity $\|\tilde{W}^T - \tilde{M}\|_F^2$ on the vertical axis. This value decreases as a function of training, indicating that the network classifier and the centered-means matrices become proportional to each other (self-duality).

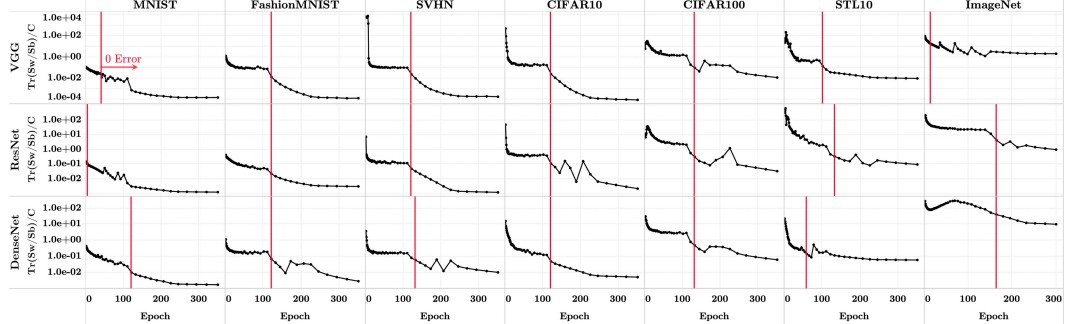

Figure 7: (ref. Figure 6 in Papyan et al. (2020)) **Training within-class variation collapses.** In each array cell, the vertical axis (log scaled) shows the magnitude of the between-class covariance compared with the within-class covariance of the train activations. Mathematically, this is represented by $\mathrm{Tr}(\Sigma_W \Sigma_B^+/C)$ where $\mathrm{Tr}(\cdot)$ s the trace operator, $\Sigma_W$ is the within-class covariance of the last-layer activations of the training data, $\Sigma_B$ is the corresponding between-class covariance, $C$ is the total number of classes, and $[\cdot]^+$ is Moore–Penrose pseudoinverse. This value decreases as a function of training—indicating collapse of within-class variation.