# OpenReview forum: "Detecting Out-of-Distribution through the Lens of Neural Collapse"
_ICLR.cc/2025/Conference — ICLR 2025 Conference Withdrawn Submission_

### Official Review · Reviewer_bncY · 2024-10-27

**Soundness:** 3
**Presentation:** 3
**Contribution:** 3
**Rating:** 5
**Confidence:** 4

**Summary:**

The authors propose an OOD scoring function, "NCI," inspired by the concept of neural collapse. Neural collapse is a phenomenon observed toward the end of training, primarily in the penultimate layer of DNNs, where features of in-distribution (ID) data align closely with the network's weights. Building on previous applications of neural collapse in OOD detection, the authors introduce a metric called pscore, defined as the angle between the ID feature and the network weights, scaled by the norm of the weights. Given that pscore is closely related to the clustering capacity of the DNN, they augment it with an additional term, the norm of the penultimate feature representation, to develop NCI as a robust OOD score.

**Strengths:**

- The method seems effective for OOD detection and is lightweight.
- Although already explored in previous works, the theoretical inspiration from Neural collapse is interesting.

**Weaknesses:**

**1/ Proof Issue in Theorem 3.1**
There may be a minor issue in the proof of Theorem 3.1 related to the alignment between class clusters and the weight vector. Specifically, it seems wrong  that \Sigma_{W} ->0 necessarily implies  (h_{i,c} - \mu_c)(h_{i,c} - \mu_c)^T \to 0(hi,c​−μc​)(hi,c​−μc​)T->0. Revisiting the proof could clarify this aspect.

**2/ UMAP Visualizations**
While UMAP provides visually appealing feature visualizations, relying on it for interpretive arguments is limited because UMAP is non-linear and does not preserve global structure or accurately reflect true density and cluster separation. To strengthen this analysis, could the authors replicate the visualizations using PCA? Additionally, it would be valuable to apply this analysis across a diverse set of DNNs and ID vs. OOD datasets, such as:
- ResNet-18: CIFAR-10 vs. CIFAR-100; ImageNet vs. ImageNet-O, Textures, iNaturalist, SUN, and Places365
- ResNet-50: CIFAR-10 vs. SVHN, CIFAR-100; ImageNet vs. ImageNet-O, Textures, iNaturalist, SUN, and Places365
- ViT: CIFAR-10 vs. SVHN, CIFAR-100; ImageNet vs. ImageNet-O, Textures, iNaturalist, SUN, and Places365

 Q/ Could the authors explain why they chose UMAP over other visualization techniques, and whether they considered alternatives like PCA.?

**3/ Hyperparameter Transparency**
For reproducibility, it would be helpful if the authors provided the hyperparameter choices for all comparison methods to ensure transparency.

Q/ could the authors include a table in the appendix and in the answer listing all the hyperparameters for each method (also their model)?

**4/ Further Collapse for Improved NCI Performance**
If NCI leverages neural collapse (NC) to detect OODs, would further convergence to NC enhance its performance? Specifically, would training the model further to achieve stronger NC improve the efficacy of NCI?

Q/ could the authors save the pscore and NCI and the NC values (1 to 4) of a DNN during the training to see how it behave for most of the DNNs in 2/ for a long training.

**5/ Transformer Model Results**
The results for transformer models are incomplete, with some methods missing. Is there a specific reason for this? A complete table with transformer model results should be included.

**6/ OpenOOD Benchmark and Covariate Shift**
The OpenOOD benchmark includes additional datasets that assess covariate shifts. Could the authors explain why these experiments were excluded? It would be insightful to test NCI’s performance under covariate shift settings and include additional results for this scenario.

**Questions:**

See most of the questions on the weakness.

---

### Official Review · Reviewer_rKeD · 2024-10-29

**Soundness:** 2
**Presentation:** 2
**Contribution:** 2
**Rating:** 5
**Confidence:** 4

**Summary:**

This paper investigates OOD detection problem through the lens of neural collapse. The authors reveal that ID features would converge to class vectors during training, while OOD features tend to cluster near the origin. Based on this observation, they introduce the NCI method, which quantifies the proximity between feature vectors and class vectors. Experiments conducted on the ImageNet and CIFAR-10 benchmarks further validate the effectiveness of NCI.

**Strengths:**

1. The motivation is sound. The perspective of neural collapse is novel.
2. The proposed NCI method is both simple and efficient for OOD detection.
3. The authors conduct experiments on multiple architectures (including ResNet-18, ResNet-50, Vit B/16, and Swin v2), showing the effectiveness of NCI.

**Weaknesses:**

1. Although NCI ranks in the top three on two benchmarks, the improvements over existing SoTA methods are marginal. Previous leading methods on single benchmarks exhibit significantly larger gains compared to NCI. This raises questions about the effectiveness.

2. While the introduction of a filtering score to enhance the separation between ID and OOD samples is reasonable, it primarily serves as a mitigation strategy for misclassifying OOD samples. A more thorough analysis of success and failure cases using NCI should be warranted, as many models may not be well-trained and could violate the neural convergence law.

3. The observation that ID features exhibit greater similarity to class vectors than OOD features is not surprising, given that OOD samples often originate from unknown classes. Thus, analyzing the effectiveness of NCI in scenarios where ID and OOD classes are similar or identical is essential. Additionally, the authors should clearly explain the distinctions between the ID and OOD data used in experiments.

4. Some claims made in the paper are overly broad. For instance, the statement, "Through the lens of Neural Collapse, we explain, connect, and complete prior methods under a holistic view" in Introduction, requires more substantiation. Given the current scope of the method discussion, it seems not possible to extend the neural collapse theory to most prior methods.

5. The organization of this paper should be improved. The introduction appears somewhat redundant with Section 3. Is it necessary to discuss the question, “Where do features of ID samples form clusters?” in both sections?

6. It would be beneficial for the authors to explore the relationship between Neuron Collapse and NCI performance, particularly in light of existing metrics for collapse quantification.

**Questions:**

How do the authors choose “CIFAR-10 Strong”/ “ImageNet Strong”? They are the top methods (e.g., top-5) in OpenOOD? Why weren’t other methods in OpenOOD included for comparison?

---

### Official Review · Reviewer_mGvB · 2024-10-31

**Soundness:** 3
**Presentation:** 3
**Contribution:** 1
**Rating:** 5
**Confidence:** 4

**Summary:**

The paper focuses on Out-of-Distribution (OOD) detection, which is important for the safe deployment of AI models. The authors propose a OOD detector inspired by the concept of Neural Collapse. They identify that in-distribution (ID) features cluster closer to their corresponding weight vectors, while OOD features are more dispersed. The proposed OOD detector leverages these insights by focusing on the proximity of features to weight vectors and utilizing feature norms for filtering OOD samples. Extensive experiments demonstrate the detector's efficiency and effectiveness, highlighting its ability to reduce generalization discrepancies while matching the latency of conventional score-based OOD detectors.

**Strengths:**

1. The motivation for designing an OOD detector from the perspective of neural collapse is technically sound. The proposed method, which leverages feature norms to filter OOD samples, is both simple and effective.

2. The paper is well-written, clear, and easy to follow.

3. The authors present empirical evidence showing that their straightforward method performs comparably to existing state-of-the-art OOD detection techniques across various benchmarks.

**Weaknesses:**

1. The observation that ID samples consistently cluster closer to the weight vectors than OOD samples is well-established in the literature [1,2,3,4] and is referred to as the familiarity hypothesis [5]. Designing OOD detectors based on feature norms has also been thoroughly explored [2,3,4]. As a result, the novelty of this paper appears to be marginal and limited.

2. The neural collapse view has also been explored for OOD detection in [6], and the comparison between this paper and those high related work [1-6] is missing.

3. It has been revealed that existing OOD detection methods is harmful for detecting other prediction failures of a model like misclassified ID data upon real-world application [7, 8, 9], and the simple softmax response baseline as the overall best performing method. Thus, the verification and discussion of the proposed method on detecting misclassified ID data is important.

Reference

[1] CSI: Novelty Detection via Contrastive Learning on Distributionally Shifted Instances. NeurIPS 2020.

[2] Understanding the Feature Norm for Out-of-Distribution Detection. ICCV 2023.

[3] Block Selection Method for Using Feature Norm in Out-of-distribution Detection. CVPR 2023.

[4] Feature Space Singularity for Out-of-Distribution Detection. Arxiv 2020.

[5] The familiarity hypothesis: Explaining the behavior of deep open set methods. Pattern Recognition 2022.

[6] NECO: NEural Collapse Based Out-of-distribution detection. ICLR 2024.

[7] ImageNet-OOD: Deciphering Modern Out-of-Distribution Detection Algorithms. ICLR 2024.

[8] OpenMix: Exploring Outlier Samples for Misclassification Detection. CVPR 2023.

[9] A Call to Reflect on Evaluation Practices for Failure Detection in Image Classification. ICLR 2023.

**Questions:**

Please refer to the weakness.

---

### Official Review · Reviewer_pQKh · 2024-11-04

**Soundness:** 2
**Presentation:** 3
**Contribution:** 2
**Rating:** 3
**Confidence:** 4

**Summary:**

The paper presents an approach to Out-of-Distribution (OOD) detection in machine learning models, leveraging the concept of Neural Collapse. The authors argue that In-Distribution (ID) samples' features cluster closer to the weight vectors of their predicted classes compared to OOD samples. They propose a detector that uses feature proximity to weight vectors and feature norms to filter OOD samples. The method is evaluated on various classification tasks and model architectures, demonstrating efficiency and effectiveness without requiring complete Neural Collapse convergence.

**Strengths:**

1. The paper's strength lies in its extensive experimentation across different network backbones, including ResNet, ViT, and Swin. This variety showcases the versatility of the NCI detector and its robustness in various model architectures.

2. The NCI detector is noted for its efficiency, with the paper claiming it matches the latency of a vanilla softmax-confidence detector. This is a significant advantage for real-world deployment where computational resources and time are critical.

3. The paper provides a view of OOD detection by connecting and completing prior methods under the lens of Neural Collapse, offering a unified perspective on a problem often tackled in isolation.

**Weaknesses:**

1. The observation that ID samples form clusters near the weight vectors while OOD samples reside separately is not new but considered basic knowledge in the field. The paper does not offer substantial novelty in this aspect, which may reduce its impact.

2. Besides, the conclusion drawn from Figure 2 that "early low cosine similarity equals no need for Neural Collapse convergence" is weak. The logical connection between early training observations and Neural Collapse is not well-established, and this represents a significant weakness in the paper's argumentation.

3. The paper could benefit from a more rigorous theoretical foundation to support its claims, especially concerning the relationship between the geometric structure of ID samples and **OOD** samples.

**Questions:**

Please see the weaknesses.

---

### Author Response · Authors · 2024-11-14
**Response to Novelty Concerns**

Thank you for your time and feedback. We would like to address the concerns about novelty.

While our methods may appear natural once explained, we strongly disagree with the assessment that our key observation—that in-distribution features consistently reside closer to the weight vector of the predicted class—exists in prior literature. None of the works cited by Reviewer mGvB ([1], [4], [5]) use weight vectors, focusing instead on feature-wise similarity alone. Additionally, [2] and [3] are norm-based methods that do not address the relationship between feature distance and weight vectors. While we acknowledge in the paper that norm-based approaches are well-established, our work offers a novel interpretation through the lens of Neural Collapse and uniquely combines both aspects under a holistic framework.

Furthermore, we note that Reviewer pQKh did not provide references to literature that substantiate the claim that our observation is “basic knowledge in the field”.  If there is true evidence of prior studies that match this approach, we would appreciate seeing those references.

[1] CSI: Novelty Detection via Contrastive Learning on Distributionally Shifted Instances. NeurIPS 2020.

[2] Understanding the Feature Norm for Out-of-Distribution Detection. ICCV 2023.

[3] Block Selection Method for Using Feature Norm in Out-of-distribution Detection. CVPR 2023.

[4] Feature Space Singularity for Out-of-Distribution Detection. Arxiv 2020.

[5] The familiarity hypothesis: Explaining the behavior of deep open set methods. Pattern Recognition 2022.

---

### Note · Authors · 2024-11-15

I have read and agree with the venue's withdrawal policy on behalf of myself and my co-authors.